# The Role of Higher Education in Shaping Essential Personality Traits for Achieving Success in Entrepreneurship in Spain

**DOI:** 10.3390/bs14030151

**Published:** 2024-02-21

**Authors:** Joaquín R. Puerta Gómez, Pedro Aceituno-Aceituno, Concepción Burgos García, Aitana González-Ortiz-de-Zárate

**Affiliations:** 1 Independent Researcher, Doctoral Programme in Law in Society, Madrid Open University (MOU), 28400 Madrid, Spain; jpuerta@taimar.com; 2 Department of Business Administration and Management and Economics, Madrid Open University (MOU), 28400 Madrid, Spain; concepcion.burgos@udima.es; 3Department of Work Sciences, Madrid Open University (MOU), 28400 Madrid, Spain; aitana.gonzalez@udima.es

**Keywords:** personality, leadership, decision making, self-confidence, fear of failure, university education, entrepreneurship, business success

## Abstract

Research on university education and its role in developing personality traits essential to achieving success in entrepreneurship is required because of the significance of entrepreneurship for advancements in the economic, social, technological, and environmental spheres. Additionally, the value of a university education in shaping an individual’s personality, and the necessity of emphasizing entrepreneurship in higher education for students to achieve real success, should be a priority in our society. Therefore, the aim of this paper is to explore how university education influences personality traits that are key to success in entrepreneurship in Spain. To achieve this objective, a qualitative methodology based on the study of 11 cases has been adopted. The results allow us to conclude that university education has a decisive influence on the development of the personality traits that integrally determine entrepreneurial success as the culmination of the final stage of the maturation process; however, a university education is not fundamental to the development of these traits. Nevertheless, entrepreneurs emphasized that the personality traits analyzed need to be reinforced explicitly in university education since they effectively positively impact the success of entrepreneurial initiatives.

## 1. Introduction

Research on university education and its role in developing personality traits essential to achieving success in entrepreneurship is required because of the significance of entrepreneurship for advancements in the economic, social, technological, and environmental spheres. Additionally, the value of a university education in shaping an individual’s personality, and the necessity of emphasizing entrepreneurship in higher education for students to achieve real success, should be a priority in our society.

Our research question was as follows: how does university education influence personality traits that are key to success in entrepreneurship in Spain?

Entrepreneurship plays a decisive role in global economic progress. The creation and development of new companies not only influences the economy of a country or region but also its social, technological, and environmental dimensions, given that the introduction of new products and services stimulates innovation and competitiveness, both of which function together as an essential engine for economic growth [1]. Therefore, it is crucial to understand the factors that contribute to entrepreneurship and the development of entrepreneurs [2]. In this sense, entrepreneurship cannot be defined as an isolated act but rather as a concatenated process that begins with the conception of the business idea, followed by the planning, execution, and, finally, consolidation of the business. Success emerges only at the end of the process, while difficulties can interrupt it at any time, resulting in failure.

Despite the abundant and diverse literature on the subject, no universal definition of entrepreneurial success has been established; however, it is clear that success requires entrepreneurial durability [3]. To measure it, some authors have used financial indicators such as revenue and profitability, as well as non-financial variables such as innovation and job creation. However, to measure entrepreneurial success, the entrepreneur’s point of view must be considered, including the motives that prompted them to start a business venture and the achievement of the personal goals set for themselves at the outset and in the subsequent stages [4]. Based on this last approach, the definition of entrepreneurial success in this study involves the perspective of entrepreneurs, which is understood as the perception of the entrepreneur regarding whether they have achieved success under their own assessment.

But who is the entrepreneur, what drives them, and what traits define them? Research in the past decade related to the theory of the personality traits of entrepreneurship has focused on answering these questions. Entrepreneurial success is the result of personal and external factors. The entrepreneur’s personality traits play a vital role because, along the way, they will encounter obstacles, circumstances, and frustrations that require character and determination. Being able to overcome all these hardships requires a strong character, which makes the difference between persevering and giving up, and this will ultimately mean the difference between success and failure.

Developing an entrepreneurial personality requires time and effort. Education, especially in its early stages, is crucial in shaping the attitudes and skills necessary for entrepreneurship [5], as it cultivates self-confidence and reduces the fear of failure [6]. Education must promote values and attitudes to increase the likelihood of success of entrepreneurs [7], moving towards quality entrepreneurship. Educational programs throughout life are essential for personal and cultural transformation, with university education being fundamental in the final stage of the personality formation process [8].

These two aspects, success in entrepreneurship and the influence of university education, need to be improved because, as highlighted in the recommendations of the Global University Entrepreneurial Spirit Students’ Survey (GUESSS), the most critical report in the world concerning university studies in entrepreneurship, it is not the number of entrepreneurial students that is decisive but rather the number of entrepreneurial students who are actually successful [9]. Some studies have addressed these two aspects, such as Listyaningsih et al. [10], Meng et al. [11], and Fuchs et al. [12]; however, there needs to be more studies on the importance of these two aspects, as shown in the literature review by Aparicio et al. [13], especially on the important aspect of developing personality traits from university education for entrepreneurial success. In this sense, Kuratko [14], starting almost 20 years ago, was already trying to reorient universities towards a new horizon, emphasizing the essential role of the professor for this purpose.

Studies have focused on the generation of entrepreneurial intentions among students through higher education, and it is true that there is an abundance of work in this regard [15,16,17,18,19,20,21,22,23,24,25,26]. Also, in studies related to entrepreneurial intention, personality traits have been included [27,28,29,30]. Recent studies highlight the concept of green entrepreneurship and uncover ways to develop it, such as the work of Alshebami et al. [31]. Similarly, but unrelated to university education, instruments have been introduced to assess specific personality traits to predict entrepreneurial personality [32,33,34,35], and models based on the Big Five personality traits (extraversion, emotional stability, openness to experience, self-awareness, and agreeableness) have even been created to estimate the probability of success as an entrepreneur [36]. Additionally, the Big five have been studied together with entrepreneurial education and entrepreneurial behavior [37]; however, there is a lack of qualitative studies focusing on the exploration of university studies and its influence on the development of personality traits that are key to success in entrepreneurship. For all these reasons, there is a research gap that demands studies on the influence of university education on the development of the personality traits that determine entrepreneurial success.

To present the result of our investigation, this article is structured as follows: in Section 2, we perform a literature review on the personality traits that have been regarded as the most important determinants of success in entrepreneurship. Subsequently, in Section 3, we explain the methodology of our research, including the methodological approach, study design, data extraction process, and analysis procedure. Then, in Section 4, we present the results of our research based on five aspects: (i) the contrast of the determinant personality traits; (ii) the means used in the development of personality traits; (iii) the appropriate pathways for the development of personality traits; (iv) the contribution of university education to the development of personality traits; and (v) a discussion with theoretical and practical implications. To wrap up, Section 5 shows the conclusions of the investigation carried out, its fundamental theoretical contribution, limitations, lines of future research, and a brief closing conclusion.

## 2. Review of the Literature

### 2.1. Theory of Personality Traits in Entrepreneurship

Research on the personality traits of entrepreneurs gained popularity in the mid-20th century with a multidisciplinary approach involving the disciplines of economics, psychology, sociology, and business administration. In the early years, around the 1980s, the diversity in research findings led scholars to conclude that there was no correlation between personality and entrepreneurship [38]. However, since the early 21st century, the study of entrepreneurial personality has gained renewed interest. The personality trait theory of entrepreneurship assumes that entrepreneurs have a unique personality with discernible psychological characteristics and that, if we were to develop a method for locating these characteristics, researchers could identify entrepreneurs among a sample of individuals [39]. Based on this theory, the prevalence of personality characteristics in entrepreneurs versus other populations is investigated, and the correlation of these characteristics with entrepreneurial performance factors such as entrepreneurial survival and growth is also analyzed [40].

In recent years, several studies have examined the role of personality in predicting entrepreneurial performance and the traits that distinguish entrepreneurs from non-entrepreneurs [41]. Several literature reviews have summarized the insights into entrepreneurial personality traits that have been made based on the available academic research [42,43,44,45,46,47,48]. For example, Alcaraz Rodríguez [49], in his research on the entrepreneurial profile, set out to identify the personal characteristics that favor success in the business world. He reviewed more than 50 studies and quoted more than 150 authors, identifying 60 relevant personality traits. Among these, creativity, initiative, self-confidence, energy, perseverance, leadership, risk acceptance, need for achievement, tolerance to change, and problem-solving skills stand out. For its part, García del Junco et al. [8] conducted their study by following successful small- and medium-sized companies for at least five years. From the literature, they took the personal determinants of success that were most frequently repeated in research studies or that were given greater relevance and studied the presence of these characteristics in the entrepreneurs of the selected sample. The results made it possible to classify the factors in order of importance, dividing them into two categories: generators and enhancers. The former are essential and are associated with creativity, tenacity, dedication, self-confidence, personal pride, and solid management skills, while the latter reinforce the entrepreneurial spirit and include ambition, the right choice of partners, the willingness to take risks, and continuous dedication.

Other authors such as Saboia Leitão and Martín Cruz [50] suggested that personality and individual qualities play a crucial role in the development and outcome of entrepreneurial initiatives and sought to discover the profile of entrepreneurs who are most likely to achieve their objectives, highlighting the importance of creativity, innovation, self-criticism, and leadership skills.

The mix of generating and enhancing factors in the personal and interpersonal dimensions determines the entrepreneurial spirit, and this, according to García del Junco et al. [8], can be measured through analyses, making it possible to identify the people who are most likely to succeed when it comes to launching an entrepreneurial project. Although this supports the idea that people’s specific characteristics influence the outcome of entrepreneurial projects [51], it is also recognized that there is no perfect profile to guarantee success in entrepreneurship [52].

In short, knowing that there is no ideal profile that brings together all the influential traits necessary for success in entrepreneurship nor one that does so in a fair way to ensure business success, we do know from its overall reiteration in the literature analyzed that entrepreneurs who have developed a high degree of self-confidence can make decisions without the fear of failure and enjoy a leadership capacity that steers them to influence and extract the best out of the people they surround themselves with, have more significant resources within themselves to take on the difficulties they encounter, and can overcome the obstacles they know they will have to deal with along the uncertain path of entrepreneurship.

### 2.2. Leadership, Self-Confidence, and Decision Making without the Fear of Failure

#### 2.2.1. Leadership

Leadership is a trait that stands out among entrepreneurs’ attributes that directly influence success [53,54]. This characteristic not only affects individual competencies but also shapes the attitude of those around the leader. Skills such as positively influencing the environment and driving capabilities towards business and personal goals are linked to this quality [55]. Leadership impacts areas such as decision making, goals, systemic thinking, proactive acting, and self-confidence. These effects are reflected in technical, interpersonal, and conceptual skills. Thus, leadership contributes to the development of entrepreneurship and facilitates the achievement of goals by channeling the efforts of collaborators [56]. Although it is related to authority, it differs in origin since leadership comes from personal skills. It is similar to power because it influences others’ decisions, but authentic leaders exercise it positively, maximizing the potential of others towards organizational objectives [55]. The entrepreneurial leader promotes progress in search of opportunities, innovation, and creative resolution by generating employee identification and commitment. The essence lies in motivating and improving the environment [55] while impacting the entrepreneur’s competencies [56].

Several studies relate leadership to entrepreneurship, such as the international exploratory study conducted by Felix et al. (2019) [57], which examined data from 34 countries and found that leadership had a determinant effect on entrepreneurship. The study by Van Hemmen et al. (2015) [58], in which they analyzed a sample drawn from 43 countries through the Global Entrepreneurship Monitor and the Global Leadership and Organizational Behavior Effectiveness projects and found that participative leadership and higher education represented the most decisive explanatory factors in explaining the variance in entrepreneurial behavior.

#### 2.2.2. Self-Confidence

A successful entrepreneur exhibits a naturally optimistic personality, responding positively to business adversities and showing high self-confidence when confronting challenges [59]. This mindset allows them to overcome the fear of failure and tackle problems with optimism by applying action strategies [60]. However, these qualities vary in intensity among individuals, and although the willingness to take risks is inherent to the profile of success, some entrepreneurs approach challenges with disbelief [61].

Optimism, self-confidence, and the ability to overcome the fear of failure are traits that directly influence the outcome of entrepreneurial initiatives [7,62,63]. From these traits arises achievement orientation, guiding decision making towards clear goals and tackling obstacles without losing control [60], which leads to achieving one’s objectives.

Research has shown that entrepreneurs are more self-confident than the non-entrepreneurial population [64]. Similarly, self-confidence influences entrepreneurial intentions [65], determines the entrepreneurial orientation of university students [66], and is related to general entrepreneurship [67], constituting a determinant of entrepreneurial success [68].

#### 2.2.3. Decision Making without the Fear of Failure

Regarding decision making and the fear of failure concerning an entrepreneur’s profile, several studies have found that the fear of failure is negatively related to entrepreneurial behavior [69,70].

This research aims to explore how university education influences three personality traits that determine success in entrepreneurship: leadership, self-confidence, and decision making without the fear of failure. This main objective is achieved by first examining the following specific goals: the importance given to each trait, how these traits were acquired and developed, the most appropriate method for their development, and the role of university education in the development of these personality traits.

Summarizing the literature review, first, there is a need for research that explores the influence of personality traits that are key to success in entrepreneurship. Second, the theory of personality traits in entrepreneurship re-emerged in the early 21st century and assumes that entrepreneurs have a unique personality with discernible psychological characteristics. Third, among these characteristics, leadership, self-confidence, and decision making without the fear of failure have been studied as the main traits of entrepreneurs.

## 3. Research and Methodology

The methodological procedure of this work consisted of three phases. The first one focused on determining the most suitable methodological approach, seeking, at all times, to guarantee that the techniques and systems to be used provide the investigation with the necessary objectivity that allows for the extrapolation of its results. The second phase focused on the exhaustive design of the study protocol, and finally, the third phase consisted of rigorously carrying out the information collection procedure and the subsequent analysis of the data obtained.

### 3.1. Methodological Approach

Based on the methodological selection criteria, an exploratory and qualitative approach is justified, highlighting its inductive nature and emphasis on understanding rather than explaining [71]. Using this qualitative approach is not about quantifying but rather comprehending a phenomenon and establishing how one aspect relates to another [72]. Answers to research questions about the how and why of a process need qualitative approaches because quantitative ones answer aspects related to what, who, where, and how much [73].

Among the qualitative techniques, the case study methodology has been chosen because of its ability to describe, understand, and analyze a phenomenon in its natural context. This exploratory technique stands out for its suitability for situations where the dividing lines between the phenomenon, its context, and the role of the researcher are not clear [74,75]. Additionally, the method meets the required validity criteria: internal validity, external validity, and reliability using multiple sources of evidence, a comparative analysis, and the protocolization of the research process [76]. Implementing this action protocol provides reliability to the results obtained through the study methodology applied. Since this involved multiple case studies, the individual observation and analysis of each case was carried out in four steps:Obtaining data and information on the case.Writing up each case.The detailed and reflective analysis of each case individually and in relation to the others.The establishment of results.

This process was carried out in two phases: the first consisted of fieldwork in which detailed and individualized information was obtained for each of the cases from the collection of previous documentation, the protocolized interview, the additional information obtained, and the contrast of various sources. The second phase comprised analyzing the data previously collected from each observation unit, drafting the case studies, and conducting their comparative analysis.

Considering the importance of maintaining a diverse perspective, the sources of information used to analyze the cases in this research encompass a variety of approaches. This includes conducting in-depth interviews, the direct observation of each case in its natural setting, reviewing related documents, and additional information obtained through third parties. The records collected were obtained from official and unofficial sources, the latter including documents generated by the entrepreneurs themselves and other individuals. This information extracted from the documents served a dual purpose: firstly, it was essential for preparing the subsequent in-depth interviews, and secondly, it allowed the data obtained through other methods, including the interviews themselves, to be contrasted.

On the other hand, in-depth interviews were the primary source of information on which the data collection and subsequent analysis were based. These interviews have pursued two fundamental objectives: Firstly, they were used for preliminary exploratory purposes, seeking to contrast, clarify, and deepen the subject of study [71]. Secondly, they had the purpose of accessing the subjective experience of the participants, overcoming the limitations imposed by the psychological and environmental context in which they are immersed [77]. The interviews aimed to address all the points and aspects studied from different angles, thus obtaining abundant and valid information for the subsequent analysis. After an initial phase in which data were obtained on the objective elements that served to outline the case, questions were asked about each of the personality traits that were decisive in the outcome of the venture, referring to the following:The importance that the interviewee assigns to each trait concerning success in the venture.How it was acquired and developed.The most appropriate way to develop these personality traits, specifically in the context of favoring success in entrepreneurship.Finally, the interviewer asked for a statement on the role higher education has in developing the personality traits that determine entrepreneurial success.

Likewise, each of the interviews was prepared meticulously following the following protocol:More than one candidate was selected for each profile in the sample.An analysis of the initial documentation of all the candidates was carried out to validate their suitability in each sample group.According to the established profiles and criteria, candidates were recruited through personal networks. None of the interviewees were known to the interviewer. Once they had been recruited, they were contacted to propose their participation in the study and to inform them of the general objectives of the investigation.A date, time, and place or mode were set for an interview lasting 60–90 min. The confidentiality considerations regarding personal data and the treatment of their information provided were explained.The interview script and a dossier with the initial and official documentation were prepared.

The interview responses were coded by three researchers, with one of them acting as a coordinator. Given that the objective of the work is based on the understanding of the phenomenon to be studied, the interview questions were open, allowing the information to flow with great freedom [78]. Therefore, the coding was open given the responses obtained following the guideline for analyzing the information necessary to achieve the general objective and the specific ones established in the study.

### 3.2. Study Design

In relation to the study’s design, various approaches have been adopted to provide it with objectivity and to ensure the extrapolation of its results. Considering the marked diversity within the entrepreneurial community, incorporating a heterogeneous sample of participants may lead to more enriching results than if a homogeneous sample were chosen [79]. This diversity reinforces the credibility and robustness of the research results.

In order to obtain a wealth of information, the cases were selected based on the following three criteria: (1) university graduates; (2) having achieved success in their entrepreneurship under their own self-assessment based on personal criteria; and (3) having a minimum entrepreneurial experience of 3 years. These participants belonged to the country of Spain, which serves as an interesting location of study because this nation needs to improve the aspects of success in entrepreneurship and university influence, as previously outlined in the GUESSS recommendations [9]. According to this report, regarding university training in entrepreneurship, 62.3% of university students in Spain have never received specific training in entrepreneurship [9,80], and this figure increases with more recent statistics from the same report (63.5%) [81,82]. Similarly, concerning university students who are in the process of launching their own businesses (nascent entrepreneurs), the percentage is also among the lowest in the GUESSS survey (18.9% compared to the average of 28.4%) [81,82].

The Global Entrepreneurship Monitor (GEM), a large-scale study conducted annually in more than 100 countries since 1999, was also used to select these cases. The GEM is an essential reference for researchers studying the phenomenon entrepreneurship. In Spain, the GEM profiles entrepreneurs according to various socioeconomic variables such as age, education level, income, and gender. To adapt this distribution to the particularities of our research and to obtain extrapolatable results, each of these characteristics was thoroughly evaluated, followed by its acceptance or rejection, and if accepted, their weighting criteria was determined.

In this sense, the distribution according to age groups was accepted, favoring groups with notable differences and over-representing those with greater representation, excluding, due to impracticality, the 18–24-year age group. To address the education variable, a specific bias was adopted: as the focus is on the influence of higher education on entrepreneurial personality traits, the sample was restricted to entrepreneurs who have received a higher education.

Regarding income levels, the approach adopted in this research does not reveal a relevant relationship between this variable and the object of study. As a result, the distribution by income was discarded. The gap between women’s and men’s participation in entrepreneurship has been closing over the last few decades. However, as shown by the Observatorio del Emprendimiento de España [83], from 2005 to the present, only the rate of female entrepreneurship was higher in the last record (5.6% compared to 5.4% for men). This gender distribution of the entrepreneurial profile must be considered when selecting the sample, not only for quantitative reasons but also qualitative reasons, since the interpretation of the phenomenon studied may be different from one to the other.

When facing the dimension of sample size, it is essential to bear in mind that the method does not involve a statistical but rather an analytical generalization [84], which is consistent with what was previously stated [82] about the need in this work to understand the phenomenon and establish how university education influences the personality traits that are determinants of success in entrepreneurship. Although the literature presents diverse perspectives regarding the optimal sample size, since Chiva Gómez [84] suggests a minimum of four cases, and Eisenhardt [85] and Yin [76] opt for between four and ten study units, it is recognized that the choice of the number of cases to study is discretionary in nature [86] and subject to the desired quality of the results, as the studies with the largest number of cases are those with the highest-quality conclusions [87].

In view of this reflection, the aims of the study, the nature of the phenomenon, the accepted criteria for segmenting the population universe, as well as their weighting, it has been decided that a study of 11 cases should be carried out since it has been considered, from a quantitative point of view, that this is a sufficient number of cases to achieve the intended objectives and, from a qualitative perspective, that they represent the diversity of the current group of entrepreneurs in Spain. The distribution of the 11 cases is structured as shown in Table 1. It follows the chronological order of the time of the in-depth interview, which is the primary source of information extracted for the study. These interviews were conducted between December 2021 and April 2022.

After establishing the distribution of the sample, the process was oriented towards the choice of the observation units. In addition to guaranteeing the desired representativeness, priority was given to the accessibility of the necessary information [88] and the willingness of individuals to participate [89]. Flexibility and willingness to collaborate were taken into account, as well as the ability of those selected to express themselves [90]. However, it was considered essential to contrast this selection criterion by consulting experts in order to support the methodological soundness of the research.

### 3.3. Data Collection and Analysis Procedure

As a result of this process, the cases were selected, and the following is a brief excerpt of the most important information on each of them in direct relation to the research objectives:Case 1. An entrepreneur in the healthcare sector has been running his organization for over 25 years after his father passed it on to him. He has negative feelings toward his university experience because he believes it did not prepare him for the business world.Case 2. A 50-year-old woman with a solid university and post-university education is starting a business in the communications and graphic arts sector after having worked as an employee. She is very satisfied with the humanistic training she received at university; however, she thinks it was not useful for entrepreneurship.Case 3. A late vocational entrepreneur who, after a long professional career as an employee, has decided, at the age of 56, to start her own technology-based quality consulting business. She describes her time at university as positive; however, it was not useful for her regarding entrepreneurship.Case 4. A young entrepreneur who started his business right after finishing his university studies is considered one of the most successful young entrepreneurs in our country today. His time at university was enriching, contributing greatly to his personal development.Case 5. An engineer who, after thirteen years of experience working as an employee, decided to start his business venture in the United States, later expanding it to other countries. He considers that his time at university was a determining factor for his subsequent entrepreneurship, as the training he received was essential for his subsequent learning.Case 6. An entrepreneur, by vocation, whose business is linked to the construction sector is at the forefront of the use of cutting-edge technology applied to this field. University helped him think and gave him various skills that increased his self-confidence, which he might not have had if he had not studied.Case 7. After three years of working in a multinational company, this entrepreneur created his own business in the communications sector. He has diversified his activities with several companies. University opened the doors to the world of employment, although it was not a determining factor in developing his entrepreneurial initiative.Case 8. This entrepreneur managed and developed a business project in the United States in the field of medical technology that has been very successful. He strongly affirms that his three university experiences were positive and relevant to his entrepreneurship, especially his time at Harvard University.Case 9. After working as an employee for several successful entrepreneurs in our country, this entrepreneur started her own business in the leisure and tourism sector. She is convinced that if she had not gone to university, she would not be the person she is today, as she would not have been able to acquire the skills required to become an entrepreneur.Case 10. This young entrepreneur began her professional career with an international perspective due, in large part, to her university education outside of Spain. Her university studies provided her with essential personal training and a knowledge base on which she later built the rest of her skills.Case 11. Passionate about artificial intelligence, this engineer has applied this technology to digital marketing, which has been the seed of her business. Her time at university was necessary to acquire the necessary technological skills required for her business project.

After coding, the data were analyzed through an iterative comparison of the developments and experiences generated in the different cases with the differences produced in the different contexts. This reflective iterative process led to a detailed understanding of the problem that allowed a global perspective of the phenomenon studied to be acquired [91].

The pattern adhered to during data analysis process was developed following the three iterative stages described by Miles et al. [92]:(a)Data reduction. The qualitative data obtained empirically were taken and synthesized in an orderly manner so that it allowed for the disaggregation of such data through sets ordered by areas, making their classification possible. This stage of the process had particular relevance in treating the information collected through the interviews.(b)Data visualization. To facilitate the interpretation of the data collected, they have been arranged through tables and matrices, as this allows for an overall view of the information related to a specific aspect, thus making it possible to compare homogeneous data, which, in the beginning, were shown in a dispersed way.(c)Verification of interpretations and preliminary conclusions. Having reached a reasoned understanding of the structured information, verifying the coherence of the interpretations made from the comparative analysis of the data obtained through the different sources was necessary. This made it possible to contrast results to obtain solid interpretations based on the preliminary results. This stage has made it possible to go deeper into the object of the study, obtaining both a global view of the problem from its different aspects and the possibility of discovering the different details and specific aspects of which the matter studied is composed.

This analysis was structured along the following lines: First, the personality traits of entrepreneurs that influence the success of their business projects highlighted by the literature were assessed, explaining why they are determinant. Secondly, how entrepreneurs have managed to develop these personality traits was evaluated. Finally, the critical factor of the university experience of the subjects under study concerning the development of these traits was estimated in an attempt to conclude whether this step—completing higher education—had a positive influence on the success of their ventures. The results of this analysis are shown in the following section.

## 4. Results

### 4.1. Contrast of the Determinant Personality Traits

To visualize this contrast, Table 2 shows the evaluation of psychological traits in relation to the determination of entrepreneurial success. The first and most relevant observation was that practically all the participants pointed out the following psychological traits, highlighting their essential relevance as a fundamental aspect in their entrepreneurial initiatives. Once this first observation was made, this set of traits was analyzed individually, and the interviewees had the opportunity to evaluate them and explain the reasons for their consideration.

#### 4.1.1. Leadership

This trait is the most highly valued by the participants out of all those included in the analysis. In this case, of the eleven members of the sample, eight considered it essential, and the remaining four considered it very important for the influence it exerts on the possibilities of the success of a venture (see Table 2). Notably, despite the different views that age and experience can give this feature, there are no significant differences between the assessments and reasons provided by young people and veterans.

Among the most senior, one (case 1) claims to have discovered the importance of leadership through experience until he became convinced that a company cannot be understood without leadership. Another very experienced entrepreneur (case 6) was emphatic about this trait, stating that if the entrepreneur cannot lead, they will not be able to convince any client, nor will they be able to drag a team along. These statements should lead to the question of whether the entrepreneur’s leadership capacity not only makes it possible to increase the probability of success of his entrepreneurial initiative but, beyond that, is necessary to avoid failure.

On the other hand, the youngest entrepreneur in the sample (case 10) was convinced that well-exercised leadership causes a transformation in the attitude of the people who collaborate with the entrepreneur, making them willing to pursue and achieve the same objectives. Two interviewees referred to the importance of this trait at the start of a business venture (case 5 and case 8).

However, exercising leadership does not only affect others. For one of the middle-aged entrepreneurs (case 11), self-leadership was essential since an entrepreneur needs to develop great inner strength to overcome the obstacles that will appear in their path, which will enable them to continue moving forward.

Finally, the most veteran entrepreneur (case 3) stated that leading is not easy at any time, and that the degree of difficulty increases if leadership has to be exercised at a distance, as has happened in a generalized manner during the COVID-19 pandemic.

It has been verified that the participants in the study have given, in general, the highest value to this feature, and that the success of their business projects hinges on it, so it can be argued that this feature influences success in entrepreneurship and, moreover, is necessary for the entrepreneur to avoid failure.

#### 4.1.2. Self-Confidence

Although leadership is the trait most valued by the entrepreneurs consulted, self-confidence follows closely behind since six of the eleven participants considered it an essential quality for entrepreneurship, and the other five thought it was very important (see Table 2). From the observations made, differences can be found between the ratings given by men and women since five of the six male entrepreneurs believed it to be an essential trait. In comparison, only one of the five female entrepreneurs rated it as such, while the other four said it was a very important quality.

However, no significant differences were found in the ratings due to age bias, as the youngest and oldest respondents agreed on the perspective on which they base their motives. The entrepreneur who has not yet reached 35 years of age (case 10) argued that entrepreneurship is associated with risk and that only through self-confidence can an entrepreneur find the courage to leave everything behind and embark on an uncertain project be nurtured. The two more experienced participants also associated self-confidence with the essence of entrepreneurship. One of them (case 1) argued that “if you don’t have confidence in what you do, you can’t be an entrepreneur (…) because entrepreneurship means risking your wealth every day”. For his part, another entrepreneur (case 5) stressed the importance of self-confidence, stating that “if the entrepreneur is not sure of himself and what he does, he cannot transmit it to others” and going so far as to recognize that if he did not start his entrepreneurship journey, it would have been because he did not feel confident, thinking that he lacked essential knowledge and skills. As a remarkable fact, three members of the sample, all of them middle-aged, associated this trait with resilience.

In view of the ratings given by the interviewees and the reasons expressed to justify them, self-confidence can be considered an influential trait in achieving success in entrepreneurship.

#### 4.1.3. Decision Making and the Fear of Failure

The fear of failure is a widespread feeling among entrepreneurs, so it is a normal situation. However, when this fear intensifies and/or becomes chronic, it could become a hindrance in the person’s decision-making process, potentially affecting the success of the venture.

As for the influence that this feature has had on the success of the businesses of the participants in this study, there is a wide dispersion, both in the assessment of the degree of relevance and its motivation as well as in the way in which they say they have dealt with it.

Among those who defend this as an essential trait, one of the interviewees (case 8) pointed out how the main blockage of brilliant people comes from not daring to make decisions and so-called paralysis by analysis. Along the same lines, another entrepreneur (case 9) stated that the fear of failure could lead to the collapse of a business project and added that, for this reason, starting a business alone is very difficult, almost impossible, since having the support of other people is essential to overcome this fear.

One of the young entrepreneurs (case 4) recognized that the fear of failure is present in his daily life, and overcoming it is vital so that it does not hinder his decision-making process. Relativizing this fear, dissociating his entrepreneurial side from his personal side, is a tool he uses that gives him excellent results.

A very interesting insight was provided by the person who acknowledged having experienced failure firsthand (case 6) since it has led him to learn about its consequences. The knowledge obtained has been essential to face the fear of not knowing how to assess his decisions. He also showed feelings of loneliness that entrepreneurs suffer when faced with failure.

Along the same lines, another participant (case 7) acknowledges feeling the fear of failure because he is aware of what is at stake every day and that the decisions he makes may change the project’s future. This entrepreneur also referred to the loneliness he has felt on many occasions. Several participants share this feeling of loneliness. One of them (case 11) recalled that having a partner was very good for her, and although she is aware of the importance of this trait, from her point of view, she does not consider it as a determining factor.

In contrast to what has been stated up to this point, several interviewees indicated that the fear of failure has not influenced the success of their business project. Some directly stated they never had this feeling (cases 2, 3, and 5), and another one (case 11) stated that she never thinks about the final success but rather about performing her task well and meeting the daily objectives, which helps her to stay away from the fear of failure.

The personal circumstances and experiences expressed by the participants do not detract from the importance that practically all of them attach to this trait, so it can be affirmed that decision making and the fear of failure are determining features that influence success in entrepreneurship.

After analyzing the personality traits, each participant was asked to indicate those they considered most important among the traits discussed and, in addition, to add any that were not analyzed but that they thought should be part of the list.

The youngest of the entrepreneurs (case 4) considered that between 70 and 80% of the key to his entrepreneur’s success lied in managing his psychology so as not to collapse at the most challenging moments nor to fall into an excess of optimism when things are going well. Another participant (case 8) emphasized that the psychological dimension of the entrepreneur is more important for a venture’s success than the technical skills and abilities they may possess. Another colleague (case 9) reinforced this approach by arguing that the skills needed for entrepreneurship can be learned if the person has the necessary attitude.

This generalized appreciation has no gender or age bias since, as one of them indicated (case 7), the path of entrepreneurship is full of obstacles. Only a person’s mental strength and a firm conviction in what they do will make it possible for them not to lose heart and to achieve their objectives.

It should be noted that the personality trait that aroused the most interest has been one that was not explicit in the relationship and that, nevertheless, many of them have been added directly or indirectly; this trait is resilience.

Entrepreneurs must show strength and flexibility in the face of challenging situations [93]; in other words, resilience, defined by these same authors as the ability to adapt flexibly to circumstances, is a crucial characteristic of entrepreneurial success. The perceived resilience of entrepreneurs is positively associated with their perception of success [94]. It leads to higher levels of self-confidence and optimism, which are also decisive factors in achieving goals [56], and there is a direct relationship between resilience and entrepreneurial survival [95]. Resilience is essential in the crucial moments of entrepreneurial and personal projects. This quality is not innate; it comes from personal experience and can be learned and cultivated [96], giving an entrepreneur the tools to adapt to new contexts and persist by using their resources and strengths [93].

Two of the participants mentioned this directly and spontaneously. In fact, the most veteran entrepreneur (case 3) stated categorically that the key to success lies in resilience, which was understood as an entrepreneur’s ability to overcome difficult situations since in entrepreneurship, there are many moments of despair and of wanting to give up, and only if the person can overcome adversity will they be able to move forward into the future and achieve their goals. Another entrepreneur (case 5), also a veteran, justified having pointed out resilience as the most crucial trait by stating that to achieve success, it is necessary to have intermediate failures, and for this reason, it is necessary to persevere to overcome them.

In addition to these two cases, several sample members indirectly refer to resilience as an essential trait since they affirm that entrepreneurship is a path full of difficulties and that the entrepreneur needs the motivation and tools to persist in the endeavor and not give up when the obstacles seem insurmountable. For example, it is worth mentioning the comment of one of the participants (case 3) who stated that “there are many more people who give up than people who fail”.

In relation to the classification of those traits that were expressly asked about, self-confidence and leadership were, by far, the most frequently chosen to be the most influential since, as indicated by one of the interviewees (case 6), they are absolutely essential qualities that must be intensely present in the person so that the entrepreneur is able to persist in the face of adversity, to have faith and believe in themselves when rational reasons to do so are lacking, and to be able to endure when others give up.

Focusing on leadership, one of the participants (case 9) pointed out that this trait ultimately a driving force. Another (case 11) clarified that it should not only be referred to regarding its influence on a group of people, but that self-leadership should also be considered as the capacity to self-manage.

Among the traits provided spontaneously in the interviews, in addition to resilience, it is worth mentioning curiosity and the eagerness to learn to move forward, as well as a good amount of passion for what one does, as these will provide the motivation and energy necessary to move forward despite the difficulties, which is, again, an indirect reference to resilience.

### 4.2. Means Used in the Development of Personality Traits

When questioned about how they developed each of the most relevant personality traits, the participants in the study indicated very different ways.

#### 4.2.1. Leadership

Although all the participants rated this trait as essential or very important, only five of them acknowledged having received training to develop their leadership skills (cases 1, 2, 3, 5, and 8). Of these five, three (cases 1, 3, and 5) did so through specific courses promoted by the company in which they worked prior to the start-up of their business, and two (cases 2 and 8) did so through postgraduate master’s degree programs in which management skills such as leadership was taught. It is worth mentioning the testimony of one the participants (case 8) who obtained a Master of Business Administration (MBA) degree at Harvard University in the USA stated that the importance of the concept of leadership is constantly stressed throughout the program, while specific training is provided on the most effective and appropriate ways to exercise it.

On the other hand, many of them believed leadership is an innate capacity of the individual that can only be developed through training and putting it into practice. Some of them even pointed out the difficulty of doing so in the classroom (cases 4, 9, and 10), and even advocated developing it through experience or the observation and imitation of role models, as pointed out by the youngest participant (case 4).

However, one of the entrepreneurs (case 8) reported that she found the group dynamics practiced at university to be very useful for understanding the principles of leadership and laying the foundations for it. This first contact enabled her to better practice and perfect this skill during her professional career.

#### 4.2.2. Self-Confidence

As for the development of this trait, none of the interviewees reported having received specific training or coaching to improve it. Only one of them (case 4), very aware of the relevance of the psychological dimension of entrepreneurship, stated that he had read a lot about related issues, such as the functioning of the mind, neuroscience, and human behavior.

It should be noted that there is no consensus as to whether this trait is innate or acquired. In this disparity of opinion, we found an age bias since four of the younger participants thought that self-confidence is an innate quality that is part of the individual’s way of being; however, the rest maintained that it is acquired through experience as the person overcomes difficulties and can look back and become aware of the achievements attained. Notwithstanding the above and the disparity, common ground was found, as the young entrepreneurs who stated that this is an innate trait declared that it can also be cultivated and trained. One of them (case 9) considered that it should be developed from all angles during the whole formative process of the person through specific work. Another of the entrepreneurs (case 10) was convinced that continuous training in this aspect is fundamental since self-confidence must be nurtured to avoid losing it.

One of the entrepreneurs (case 6) stated that acquiring the knowledge and skills necessary for entrepreneurship is essential for gaining self-confidence, which relates the development of this trait through education. Along these lines, another young entrepreneur (case 4) described what happened at his university with his final degree project. After presenting it in public in front of an examining board made up of professors and lecturers, they were interested in asking him questions, and finally, he was congratulated and awarded the highest grade. He said this “was encouraging,” made him feel valued, and greatly reinforced his self-confidence. Related to this anecdote and the role that training can play in improving this quality, the youngest entrepreneurs (case 10) said that being surrounded by a supportive environment, both in the training process and during entrepreneurship, is a great help.

#### 4.2.3. Decision Making and Fear of Failure

Very few reported having developed this trait (cases 4, 7, and 8) because, beyond those who do not acknowledge having feared failure (cases 1, 2, 3, 5, and 10) as it is part of their personality, a fairly generalized opinion is that this feeling is overcome with experience and with having a clear objective to achieve.

A detail that cannot be overlooked is in line with the loneliness experienced by entrepreneurs that was highlighted earlier by two of the participants (cases 6 and 9) since the way in which several of the interviewees stated that they have overcome the fear of failure is to share it with people who can understand it, i.e., partners or other entrepreneurs (cases 4, 7, 9, and 11).

However, some of the participants did think that this trait can be developed. First of all, entrepreneurs must inform themselves to know the risks involved and accept them as part of life, relativizing their importance, as stated by one of the participants (case 5). Another participant (case 6) added to this idea, stating that, on the one hand, the psychological and emotional effects of failure must also be known, and on the other hand, so do the material and legal consequences involved. This entrepreneur criticized the fact that, especially in the academic world, only success stories are discussed, and failure is never dealt with.

The ability to make decisions can be exercised like a muscle according to one of the interviewees (case 8), and after years as an entrepreneur training this muscle, he claims to have this ability much more developed. Finally, and in relation to university education, one of the entrepreneurs (case 11) affirmed that facing the demanding level of studying required for engineering and successfully graduating makes the fear of failure bearable and does not interfere in his decision making.

### 4.3. Appropriate Pathways for the Development of Personality Traits

Immediately after expressing how they developed these factors, they were asked about what, in their opinion, would be the most appropriate way to develop these personality traits oriented to favor success in entrepreneurship. In response to this question, the interviewees pointed to a wide variety of avenues, with university education occupying a prominent place.

Training is essential for the development of this type of trait. This statement made by one of the interviewees (case 2) was shared by most of the sample members; however, each of them contributed their own nuance, although many agreed that this training should be ultimately practical since training an entrepreneur implies preparing them for what their will encounter later on.

Likewise, several of them believed that an entrepreneur must remain in a process of continuous learning (cases 6, 7, and 8) so that the development of these traits must be constant, and that the person must ensure their lifelong training. In this formative process, university education plays a fundamental role since it is the moment in which many of the personality traits that had been cultivated in previous stages develop and flourish, as expressed by one of the participant’s opinions (case 11).

For the development of these intrinsic aspects of a person, as pointed out by another entrepreneur (case 4), the subjects taught at university are not as relevant as the dynamics, i.e., the way in which learning is carried out. In addition to offering a more practical component, added another participant (case 7), it should reinforce the psychological training of the individual, determine the personality type of each individual, and guide them to be able to overcome the difficulties that will arise in later stages.

However, university is not the only educational channel for developing these types of factors since there were several opinions in favor of starting this process at earlier stages (cases 1, 2, 8, and 11), even as early as kindergarten, as one of them pointed out (case 1).

Although there is a consensus on the desirability of having a training base, some were convinced that the most effective way of learning is through the transmission of experience from people who have traveled the path of entrepreneurship (case 10).

Finally, there were those in favor of developing these personality traits in a way that is different from traditional training, referring mainly to the development of psychological aspects because, in their view, it requires getting to know oneself, and this can be achieved through reading and interacting with other people who can transmit, to the entrepreneur, the stages they will go through at an emotional level throughout the learning process (case 4).

The comparative analysis of the influence of an entrepreneur’s personality traits on the success of entrepreneurial projects has allowed us to observe that the selected sample generally considered that having developed and cultivated them has considerably increased their chances of success. They also agreed that training is the most appropriate way to develop these aspects. In addition, it is essential to specify that a person’s strengthening must be continuous, and that it starts being forged from an early age and is continuously reinforced throughout their maturation process.

### 4.4. Contribution of University Education to the Development of Personality Traits

The reflections provided by the participants on the contribution of university education to the development of personality traits that influence success in entrepreneurship pointed out that university plays a determining role in this process, as it usually coincides with an essential period in the development of the individual since it is aligned with the culmination of the refinement of factors that have been cultivated in previous stages.

The experience expressed by most of the subjects of the case studies showed that their university education contributed decisively to their training from an integral point of view; however, it was not essential in the development of the traits analyzed. From their entrepreneurial experience, they pointed out that these aspects should be trained during the university stage, making special mention of the reinforcement of the psychological traits and the transmission of the experience lived by people who have been on the path of entrepreneurship themselves.

### 4.5. Discussion

#### 4.5.1. Theoretical Implications

To achieve the above results, a comparative analysis of the cases studied was applied with a structure that began by assessing the personality traits of entrepreneurs highlighted by the literature that influence the success of entrepreneurial projects, explaining the reasons why they are determinant. In this aspect, the most valued traits were leadership and self-confidence. These results are in line with what has been previously uncovered in the theory of personality traits in entrepreneurship, particularly when entrepreneurship is related to leadership [53,54,55,56,57,58] and self-confidence [7,62,63,64,65,66,67,68]. Similarly, although decision making without the fear of failure obtained a lower rating, it is also considered a relevant personality trait, which aligns with the theory relating it to entrepreneurship [69,70]. In these evaluations, no age differences were found. However, they appear in the assessment of sex in relation to self-confidence as a determining factor in success in entrepreneurship, showing that they are essential in the opinion of men and only very important according to women.

However, it is interesting to note that the most exciting personality trait for the participants was one that was not explicit in the relationship and that many of them added themselves either directly or indirectly; this trait is resilience. In addition, others, such as curiosity and the eagerness to learn and to move forward, were also highlighted, as well as a good amount of passion for what one undertakes. These results were interesting, and we did not anticipate them based on the literature review. In any case, what all the participants agreed on, without finding any bias due to age or gender, is the relevance of the psychological dimension of the entrepreneur in the outcome of a venture.

Subsequently, in the comparative analysis of cases, the way entrepreneurs have developed these personality traits was evaluated. This evaluation showed a coincidence in all cases, pointing to training as an essential way to develop this set of qualities. To this point, it was added that training must result from a continuous process that should begin at an early age and be reinforced throughout a person’s maturation process.

In this formative process referred to by the sample members, university was identified as a fundamental actor in the development of their personality. Subsequently, and as a final part of the analysis, the influence of university education on the development of personality traits related to success in entrepreneurship was estimated. In this aspect, the results obtained indicate that university education contributed decisively to the growth of the personality traits studied from an integral point of view since it coincided with the final stage of their maturation process and that it was through university education that these personality traits, which were still developing in previous stages, were nurtured. Still, university education is not fundamental in the development of these traits, which is why the entrepreneurs pointed out that the personality traits analyzed should be explicitly reinforced in the university stage since they do indeed have a favorable impact on the success of entrepreneurial initiatives.

Moreover, this study differs from others which focused on the generation of entrepreneurial intentions among students through higher education [15,16,17,18,19,20,21,22,23,24,25,26], as it is oriented towards the new horizons highlighted by Kuratko [11], which are related to the fact that what is really important is that the number of entrepreneurial students who actually succeed, which is high [9]. For the same reason, this study differs from others investigating personality traits [27,28,29,30,65] but only in their relationship with entrepreneurial intention. Also, this study differs by its inclusion of university education compared to other studies, which tend to focus on instruments to assess these specific personality traits to predict entrepreneurial personality [32,33,34,35] or that have created models based on the Big Five personality traits (extraversion, emotional stability, openness to experience, consciousness, and agreeableness) to estimate the probability of success as an entrepreneur [36].

#### 4.5.2. Practical Implications

At the organizational level, our findings can inform practice by providing empirical proof of the influence of university studies on the development of personality traits present in entrepreneurs. Stakeholders such as postsecondary institutions could use these results to adjust their curriculums and to include developmental activities targeting these traits. Additionally, postsecondary institutions could actively work on the development of these traits through their mentoring programs, using developmental relationships to develop the skills, abilities, and profiles of future entrepreneurs. At the group level, these findings can also encourage professors to actively work towards the development of personality traits present in entrepreneurs so that they can contribute to the creation of a new generation of successful entrepreneurs.

In addition, our study intends to go beyond its initial objectives, seeking to position, based on its findings, the higher education system as the epicenter of the development of individuals who will constitute a solid foundation for the business sector, and highlighting the importance of strengthening essential traits that will prevent business failure, especially psychological traits that strengthen character and foster resilience. In this way, the study carried out modestly aimed to turn the Spanish university into a point of reference, promoting entrepreneurial vocations and improving the quality of entrepreneurship.

## 5. Conclusions

Considering all of the above, this work has contributed to the research gap that exists concerning the influence of university education on the development of determinant personality traits that influence entrepreneurial success by obtaining qualitative empirical data in support of the theory of entrepreneurial personality traits [39], ultimately identifying leadership, self-confidence, and decision making by controlling the fear of failure as discernible psychological characteristics.

This exploratory research has sought to understand the relationship between university education and the psychological traits that favor successful entrepreneurship, and although the results on which these conclusions are based are consistent with the understanding of this relationship, the study has the limitation of having been based on qualitative methodologies which, justified by the achievement of an analytical generalization, will have to be complemented in future works with other approaches to be able to generalize the statistical findings, quantifying the relationships between training and personality traits determining success in entrepreneurship.

Future research could be articulated on the basis of providing training to university graduates who have created companies in relation to the variables discussed in this work (leadership, self-confidence, and decision making without the fear of failure). Subsequently, future research would need to quantify the relationship between training and these personality traits that determine success in entrepreneurship before and after training. With these quantitative studies, it would be possible to observe the effects of training on these traits determining success in entrepreneurship and the potential differences that might exist between different types of characteristics of the population, such as age, sex, and the degree of success obtained so far in a business project that is underway.

Although finding cases of business failure is complicated, it would also be interesting as a future line of research to approach this through a comparison of this case type with cases that have been successful to understand the phenomenon of whether university could have helped to avoid failure and how it can help to do so in the future.

Our study shows that university education has a decisive influence on the development of the personality traits that determine entrepreneurial success in an integral way as the culmination of the final stage of the maturation process, although it is not fundamental in the development of these traits. The findings presented in this work pave the road for future studies to further explore, from the perspective of psychology, the understanding that the reinforcement of university education can specifically influence the relationship between personality traits and success in entrepreneurship.

## Figures and Tables

**Table 1 behavsci-14-00151-t001:** Distribution of case studies.

Case Number (*)	Identifier	Sex	Age Range
1	H 55–64	Man	55–64
2	M 45–54	Woman	45–54
3	M 55–64	Woman	55–64
4	H 25–34	Man	25–34
5	H 45–54	Man	45–54
6	H 45–54	Man	45–54
7	H 35–44	Man	35–44
8	H 35–44	Man	35–44
9	M 35–44	Woman	35–44
10	M 25–34	Woman	25–34
11	M 35–44	Woman	35–44

(*) Interview duration 60–90 min.

**Table 2 behavsci-14-00151-t002:** Assessment of psychological traits in relation to the determination of success in entrepreneurship.

Case No.	Identifier	Leadership	Self-Confidence	Decision Making and Fear of Failure
1	H 55–64	VI	E	VI
2	M 45–54	E	E	RI
3	M 55–64	E	VI	RI
4	H 25–34	E	E	E
5	H 45–54	VI	E	I
6	H 45–54	E	E	I
7	H 35–44	E	E	VI
8	H 35–44	E	VI	E
9	M 35–44	E	VI	I
10	M 25–34	VI	VI	VI
11	M 35–44	E	VI	I

E—essential; VI—very important; I—important; RI—relative importance.

## Data Availability

The data presented in this study are available upon request from the corresponding author.

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
