# Peer review of "The Role of Higher Education in Shaping Essential Personality Traits for Achieving Success in Entrepreneurship in Spain"

_behavsci, 2024, doi:10.3390/bs14030151_

Round 1
Reviewer 1 Report (New Reviewer)
Comments and Suggestions for Authors
The paper tackles and interesting and long-standing issue of entrepreneurial success. There are some improvements needed before the paper can be ready.
1. What is the definition of entrepreneurial success in this study?
2. What is the problem with success in the context that warrants a qualitative not a quantitative study?
3. Why focus on personality traits?
4. Why and how were the 11 cases decided upon?
5. How was the analysis done and how were the themes derived?
6. Wouldn’t it be good to identify some successful entrepreneurs and some not so successful to compare what are the differences?
7. What was found interesting and different from what is already available in the literature?
8. How can the findings inform practice and which stakeholders would the findings be targeted at?
9. The literature are mostly 2022 or prior to that now it is 2023 maybe some updating is required.
Author Response
The paper tackles and interesting and long-standing issue of entrepreneurial success. There are some improvements needed before the paper can be ready.
- What is the definition of entrepreneurial success in this study?
The definition has been provided in lines 52-55.
- What is the problem with success in the context that warrants a qualitative not a quantitative study?
We have presented the arguments used to select a qualitative approach in lines 212-220.
- Why focus on personality traits?
The relevance of the focus on personality traits has been explained in lines 68-73. Additionally, it is explained in the literature review section, under the subheading Theory of personality traits in entrepreneurship.
- Why and how were the 11 cases decided upon?
We are grateful to the reviewer for clarifying this issue. This aspect is dealt with in the article between lines 304-316. Quantitatively, we exceed the number of cases required by the expert authors in this field by 1 (11 out of 10) and qualitatively, we meet the objective of analytical generalisation, and we consider that a sample as representative as possible can offer a greater wealth of information, which serves to better understand the phenomenon studied.
- How was the analysis done and how were the themes derived?
We are also grateful to the reviewer for clarifying this aspect. Given the research gap described in lines 80-88 in relation to the lack of studies on the influence of the university on the success of student entrepreneurs, we have reviewed the existing studies on this subject and the most reiterated in the literature analysed are those on which our team has carried out the study as described in lines 151-156: leadership, self-confidence, and decision-making without fear of failure.
- Wouldn't it be good to identify some successful entrepreneurs and some not so successful to compare what are the differences?
We find it an interesting study, because negative cases provide a wealth of information and it feeds into future lines of research in lines 705-707. In this study, we have tried to follow the recommendations made by the global GUESSS Project to universities and public institutions, when it expresses that it is not the number of entrepreneurial students that is decisive, but the number of entrepreneurial students who succeed [9] (lines 81-84).
- What was found interesting and different from what is already available in the literature?
Please refer to lines 662-670.
- How can the findings inform practice and which stakeholders would the findings be targeted at?
We have addressed this in lines 671-679.
- The literature are mostly 2022 or prior to that now it is 2023 maybe some updating is required.
We have included more updated literature. Examples:
10. Listyaningsih, E.; Mufahamah, E.; Mukminin, A.; Ibarra, F. P.; Santos, M. R. H. M. D.; Quicho, R. F. Entrepreneurship education, entrepreneurship intentions, and entrepreneurship motivation on students’ entrepreneurship interest in entre-preneurship among higher education students. Power Educ 2023, 17577438231217035.
11. Meng, D., Shang, Y., Zhang, X., & Li, Y. (2023). Does Entrepreneurship Policy Encourage College Graduates’ Entrepreneurship Behavior: The Intermediary Role Based on Entrepreneurship Willingness. Sustainability, 15(12), 9492.
12. Fuchs, L., Bombaerts, G., & Reymen, I. (2023). Does entrepreneurship belong in the academy? Revisiting the idea of the university. Journal of Responsible Innovation, 1-19.
31. Alshebami, A. S.; Alholiby, M. S.; Elshaer, I. A.; Sobaih, A. E. E.; Al Marri, S. H. Examining the Relationship between Green Mindfulness, Spiritual Intelligence, and Environmental Self Identity: Unveiling the Path to Green Entrepreneurial Intention. Administrative Sci 2023, 13(10), 226.
37. Ouni, S.; Boujelbene, Y. The mediating role of big five traits and self-efficacy on the relationship between entrepreneur-ship education and entrepreneurial behavior: study of Tunisian university graduate employees. Eval Program Plann 2023, 102325.
Reviewer 2 Report (New Reviewer)
Comments and Suggestions for Authors
Thank you for the opportunity to evaluate this article. Kindly note the following suggestions:
1. Revise to read as follows: "The role of higher education in shaping essential personality traits for achieving success in entrepreneurship."
2. A thorough proofreading is needed to address grammatical errors in the article.
3. It is crucial to separate the theoretical and practical implications into distinct sections for the benefit of both the study and readers.
4. Provide a more comprehensive conclusion summarizing the entire article from start to finish.
5. In the abstract, the statement "The aim of this paper is to explore how training influences personality traits that are key to success in entrepreneurship" contradicts the later assertion, "The results obtained allow us to conclude that university education has a decisive influence." Clarify whether the focus is on training or university education in the abstract.
6. Although the study is qualitative, statements like "direct and indirect relationship" may be inappropriate. Ensure the language used aligns with the qualitative nature of the research.
7. Clearly highlight the research gap, especially considering existing studies on education, personality traits, and entrepreneurship. Strengthen the foundation of the study.
8. Refer to recent studies, such as "Examining the Relationship between Green Mindfulness, Spiritual Intelligence, and Environmental Self Identity: Unveiling the Path to Green Entrepreneurial Intention, 2023," to enrich the literature review.
9. Replace "Material and methods" with "Research methodology," which is more suitable for management and social sciences.
10. Conclude the introduction with a section outlining the organization of the study.
11. Due to the article's extensive details, consider dividing it into specific sections with clear distinctions for the study variables.
12. Despite being a qualitative study, consider creating a visual representation of the study model to enhance clarity. The current description is confusing and a visual aid could improve understanding.
13. Split the research methodology section into distinct parts.
Best of luck.
Comments on the Quality of English Languageneeds improvement
Author Response
Thank you for the opportunity to evaluate this article. Kindly note the following suggestions:
- Revise to read as follows: "The role of higher education in shaping essential personality traits for achieving success in entrepreneurship".
This has been modified.
- A thorough proofreading is needed to address grammatical errors in the article.
We have proofread the complete manuscript.
- It is crucial to separate the theoretical and practical implications into distinct sections for the benefit of both the study and readers.
We have split the discussion in two: theoretical and practical implications, as recommended.
- Provide a more comprehensive conclusion summarizing the entire article from start to finish.
The conclusion has been expanded and the main summary of the article has been included.
- In the abstract, the statement "The aim of this paper is to explore how training influences personality traits that are key to success in entrepreneurship" contradicts the later assertion, "The results obtained allow us to conclude that university education has a decisive influence". Clarify whether the focus is on training or university education in the abstract.
We have clarified this aspect using the term university studies in both sentences.
- Although the study is qualitative, statements like "direct and indirect relationship" may be inappropriate. Ensure the language used aligns with the qualitative nature of the research.
We are grateful to the reviewer for this clarification in order to be more polite with the language of qualitative research and we have removed the statements relating to "direct and indirect relationship" in the corresponding paragraphs in the following lines: abstract (31-39), (621-622), (659-661) and conclusions (714-718).
- Clearly highlight the research gap, especially considering existing studies on education, personality traits, and entrepreneurship. Strengthen the foundation of the study.
More effort has been placed on clearly highlighting the research gap (lines 92-103).
- Refer to recent studies, such as "Examining the Relationship between Green Mindfulness, Spiritual Intelligence, and Environmental Self Identity: Unveiling the Path to Green Entrepreneurial Intention, 2023," to enrich the literature review.
The study has been mentioned in the introduction (lines 94-95).
- Replace "Material and methods" with "Research methodology," which is more suitable for management and social sciences.
Done (line 204).
- Conclude the introduction with a section outlining the organisation of the study.
This has been included (lines 104-113).
- Due to the article's extensive details, consider dividing it into specific sections with clear distinctions for the study variables.
We find your suggestion very helpful, as the division of the article into sections makes it easier to read. In order to enable the reader to follow the research process and its results and conclusions in a clear and structured way, a paragraph explaining the outline of the article has been added at the end of the first section (lines 104-112).
- Despite being a qualitative study, consider creating a visual representation of the study model to enhance clarity. The current description is confusing and a visual aid could improve understanding.
We are grateful that the reviewer has helped us to perceive the need for a visual representation of the study model in order to make it easier to understand and follow throughout the article. To this end, at the beginning of section 3 we have included an explanatory introduction and a figure that graphically represents the three phases into which the study methodology is structured, as well as the steps into which each of them is divided.
- Split the research methodology section into distinct parts.
Following the revision suggested in the previous point, the methodology section has been divided into three different sub-sections according to the graphic scheme at the head of the section.
Reviewer 3 Report (New Reviewer)
Comments and Suggestions for Authors
Many thanks for your submission to the Behavioural Sciences journal.
I believe that the manuscript is an interesting read. However, please see comments below which I hope would go towards strengthening the manuscript.
· I believe, from the outset, that the title could provide further insight into the type of university education and confirm the context (where the study was conducted).
· I would suggest that the first sentence of the abstract is rewritten. At present, it contains too many words and is general, in highlighting the research problem.
· Where is the study being conducted? The abstract should ideally allude to this.
· The opening sentences of section, such as the introduction, should refer to sources to evidence wider reading and confirming the background to the study and the overarching research problem.
· Ensure that the closing paragraph of the introduction contains a brief description of the contents of the manuscript, section-by-section.
· The literature review section opens with some bold questions. I think that these should be introduced, and their importance explained, in the introduction. In my opinion, they form part of the focus of the manuscript.
· I think that a lot of theory is trying to be covered in section 2.2. This should be refined to sub-sections of key themes and elements of personality theory. Therefore, self-confidence, and decision-making may possibly open this section. Then, discussion of ideas of subsequent leadership and managerial operations may follow.
· Ensure that the closing paragraphs of the literature review provides some clear assumptions and themes being deduced from the literature. This then provides an adequate bridge from the literature review to the methodology.
· At present, section 3 is too large and requires sub-sections. Ensure that you highlight relevant studies in this subject area (the research field of enterprise and university education) and describe which methods have been utilised and for what purpose/focus. How do these influence your methodological profile and adoption of method(s)? Also, sections on the context and sampling technique should be included to provide adequate pauses in the methodology. At present, the process is somewhat hidden within a large section.
· I believe that the results (section 4) are presented well. However, the discussion and acknowledgement of sources (section 5) could be integrated with the detailed results section sooner. This may allow more space in the latter sections to highlight the value and contributions of the manuscript. This would also possibly allow the conclusion section to be bigger and to highlight recommendations for both theory and practice.
Author Response
Many thanks for your submission to the Behavioural Sciences journal.
I believe that the manuscript is an interesting read. However, please see comments below which I hope would go towards strengthening the manuscript.
- I believe, from the outset, that the title could provide further insight into the type of university education and confirm the context (where the study was conducted).
The context of the study has been included in the title (Spain).
- I would suggest that the first sentence of the abstract is rewritten. At present, it contains too many words and is general, in highlighting the research problem.
The first sentence of the abstract has been rewritten to highlight the research problem.
- Where is the study being conducted? The abstract should ideally allude to this.
We have included the context of the study in the title and abstract.
- The opening sentences of section, such as the introduction, should refer to sources to evidence wider reading and confirming the background to the study and the overarching research problem.
The opening sentences of the introduction have been modified to refer to sources to evidence wider reading and confirming the background of the study and the main research question (lines 45-51).
- Ensure that the closing paragraph of the introduction contains a brief description of the contents of the manuscript, section-by-section.
A brief description of the contents of the manuscript has been included in the closing paragraph of the introduction (lines 104-112).
- The literature review section opens with some bold questions. I think that these should be introduced, and their importance explained, in the introduction. In my opinion, they form part of the focus of the manuscript.
The bold questions have been moved to the introduction to help explain the focus of the manuscript.
- I think that a lot of theory is trying to be covered in section 2.2. This should be refined to sub-sections of key themes and elements of personality theory. Therefore, self-confidence, and decision-making may possibly open this section. Then, discussion of ideas of subsequent leadership and managerial operations may follow.
We have included three subsections: leadership, self-confidence, and decision-making.
- Ensure that the closing paragraphs of the literature review provides some clear assumptions and themes being deduced from the literature. This then provides an adequate bridge from the literature review to the methodology.
A closing paragraph of the literature review has been included summarising the main points, clearing assumptions ad themes being deduced from the literature (lines 198-202).
- At present, section 3 is too large and requires sub-sections. Ensure that you highlight relevant studies in this subject area (the research field of enterprise and university education) and describe which methods have been utilised and for what purpose/focus. How do these influence your methodological profile and adoption of method(s)? Also, sections on the context and sampling technique should be included to provide adequate pauses in the methodology. At present, the process is somewhat hidden within a large section.
It is true that the paper has taken on a large dimension that needs to be structured, for which we thank the reviewer for this comment. This aspect has already been discussed above in point 11 of reviewer 2. The qualitative approach of our study stems from the research gap highlighted both by Aparicio et al. (2019), Kuratko (2005), Listyaningsih et al. (2023), Meng et al. (2023), or Fuchs et al. (2023) and by the experts of the GUESSS project to universities and public institutions, when it expresses that it is not the number of entrepreneurial students that is decisive, but the number of entrepreneurial students who succeed [9] (lines 80-88). Therefore, we have tried to understand the phenomenon of the influence of university education on the development of personality traits that determine entrepreneurial success, and for this purpose qualitative methodology is the most appropriate. Likewise, as we have stated in future research, we would try to carry out research that quantifies the relationships between these variables.
- I believe that the results (section 4) are presented well. However, the discussion and acknowledgement of sources (section 5) could be integrated with the detailed results section sooner. This may allow more space in the latter sections to highlight the value and contributions of the manuscript. This would also possibly allow the conclusion section to be bigger and to highlight recommendations for both theory and practice.
We are grateful to the reviewer for his suggestion to highlight the usefulness of the research carried out, and to address this comment, point 5 Discussion has become point 4.5, under results. Likewise, following the reviewer's indications, the conclusions section has been expanded to include the fundamental theoretical contribution, limitations, lines of future research and a brief closing conclusion.
We finally thank this Reviewer for all her suggested improvements; we expect our modifications to plenty fulfill the Reviewers' requests.
With our best regards,
Round 2
Reviewer 2 Report (New Reviewer)
Comments and Suggestions for Authors
satisfied
Author Response
we thank the reviewer for all the contributions to improve our article.
Reviewer 3 Report (New Reviewer)
Comments and Suggestions for Authors
Many thanks for your resubmission to the Behavioural Sciences journal.
I am happy with the recent revisions and changes to the manuscript. These directly respond to comments provided in the previous review round.
I therefore recommend an accept decision, at this stage.
Author Response
Likewise. thank you very much for all your contributions to enrich our work.
This manuscript is a resubmission of an earlier submission. The following is a list of the peer review reports and author responses from that submission.
Round 1
Reviewer 1 Report
Comments and Suggestions for Authors
I am an expert on entrepreneurship more than behavioral sciences, so my comment is related to the entrepreneruship part.
Overall, the paper does not add very much to the existing literature. Also the role of resilence is well known in the entrepreneurship studies. What could be interesting (and it has not been explored) is the difference that seems to emerge from the answers of the case studies analyzed.
In fact, the younger interviewees seem to agree on a positive opinion of their university career, in some cases supporting its usefulness with respect to entrepreneurship training. This could also be important to evaluate the dramatic changes that have occurred within the university, both from the point of view of curricula and from the point of view of teaching methods. This aspect, however, is totally unexplored, making the analysis, in my opinion, incomplete.
Reviewer 2 Report
Comments and Suggestions for Authors
Dear Authors,
I want to begin by congratulating you on your work.
The topic is interesting and topical, which makes the article relevant.
The methodology seems suitable for the type of research being carried out.
To improve the article, I would suggest the following:
In Table 1 - add the duration of the interviews;
In Table 2 - change the asterisks to E, VI, I, RI and NVI.
In the last section, divide the discussion of results and conclusions.
Include a section for limitations of the study and suggestions for future research.
Best of luck with your article.
Kind Regards.
Reviewer 3 Report
Comments and Suggestions for Authors
The paper has good potential to contribute to the literature on the relationship between university education and personality traits related to leadership, self-confidence, and decision-making. The study design is sound, with a robust literature review in the introduction and a clear discussion of the rationales for the research methodologies. The results section is comprehensive, but there are some areas in which the paper could be improved.
1. The introduction and discussion sections lack theoretical grounds. It would benefit the authors to provide more background information on the theoretical frameworks guiding their study. For example, they could discuss the psychological theories linking university education to developing these personality traits. Additionally, in the discussion section, the authors should go beyond reporting their results and discuss the “so what” question, exploring the theoretical implications of their findings and how they fit into the existing body of knowledge in this area.
2. The results section contains excessive detail that can be streamlined for better readability. Some descriptive data and minor findings could be moved to appendices to avoid cluttering the main text and allow the reader to focus on the key takeaways. This will also conform to the reporting standards in this area of research.
3. The section on limitations could be expanded. The authors could discuss factors that might have influenced their results, such as the small sample size and possible biases introduced using case studies. They could also highlight the need for future research to replicate their findings with a larger sample and explore other factors that may affect the relationship between university education and personality traits.
Reviewer 4 Report
Comments and Suggestions for Authors
Thank you for the invitation to review the article titled “University Education in the Development of Personality Traits that Determine Entrepreneurial Success.” The primary aim of this research is to investigate how training influences personality traits critical to entrepreneurial success. The authors employ a qualitative methodology, based on the study of 11 entrepreneurs. I trust you will find these comments valuable.
It is advisable to rework the Introduction section of the paper. Rather than discussing the lack of definition for entrepreneurial success and failure, focus on how your research advances our understanding of this topic, including research gaps, existing knowledge, and areas that remain unexplored.
I suggest condensing the introduction and adding a literature review section before the Materials and Methods. In the literature review, introduce and discuss three key personality traits - leadership, self-confidence, and fearlessness in decision-making - identified as determinants of entrepreneurial success in your study.
The Discussion and Conclusions need enhancement. Compare the study results with previous research in different contexts, highlighting both similarities and differences.
Clearly articulate the contributions of your research to the entrepreneurship literature.
Include a section on the study's limitations and provide recommendations for future research in this area at the end of the paper.
I trust that these suggestions will aid in further refining your research paper.
Reviewer 5 Report
Comments and Suggestions for Authors
The manuscript is based on the known facts. There are no specific objectives. There is no critical write-up. Thus, the literature review lacks many significant parts to justify the purpose of the evaluation. The methodology is not complete. I cannot see any specific implication of the review/interviews. I strongly suggest the authors rewrite the manuscript prudently and reflect the journal's theme.
Comments on the Quality of English LanguageAverage